# Influence of Glycerophosphate Salt Solubility on the Gelation Mechanism of Colloidal Chitosan Systems

**DOI:** 10.3390/ijms22084043

**Published:** 2021-04-14

**Authors:** Piotr Owczarz, Anna Rył, Jarosław Sowiński

**Affiliations:** Department of Chemical Engineering, Lodz University of Technology, 90-924 Lodz, Poland; piotr.owczarz@p.lodz.pl (P.O.); jaroslaw.sowinski@p.lodz.pl (J.S.)

**Keywords:** chitosan hydrogels, glycerophosphate salt, calcium, magnesium, solubility, stability, thermoinduced aggregation, rheometry, turbidymetry

## Abstract

Recently, thermosensitive chitosan systems have attracted the interest of many researchers due to their growing application potential. Nevertheless, the mechanism of the sol-gel phase transition is still being discussed, and the glycerophosphate salt role is ambiguous. The aim of the work is to analyze the possibility of the exclusive use of a non-sodium glycerophosphate salt and to determine its impact on the gelation conditions determined by rheological and turbidimetric measurements as well as the stability of the systems by measuring changes in the Zeta potential value. It was found that ensuring the same proportions of glycerophosphate ions differing in cation to amino groups present in chitosan chains, leads to obtaining systems significantly different in viscoelastic properties and phase transition conditions. It was clearly shown that the systems with the calcium glycerophosphate, the insoluble form of which may constitute additional aggregation nuclei, undergo the gelation the fastest. The use of magnesium glycerophosphate salt delays the gelation due to the heat-induced dissolution of the salt. Thus, it was unequivocally demonstrated that the formulation of the gelation mechanism of thermosensitive chitosan systems based solely on the concentration of glycerophosphate without discussing its type is incorrect.

## 1. Introduction

In the last few decades, there has been a growing interest in the development of smart polymer systems [1] exhibiting a sharp change in properties in response to chemical and/or physical factors [2]. The most frequently used stimulus used in biomedical application, both in in vitro and in vivo tests, is temperature, due to the relatively easy way to control it during tests as well as mapping physiological conditions. An example of systems characterized by the lower critical solution temperature [3], i.e., the critical temperature below which the polymer-solvent system is single-phase, and above which phase separation occurs [4], are colloidal chitosan systems obtained in accordance with the methodology proposed by Chenite et al. [5,6]. This solution is based on dissolving the polymer in a hydrochloric acid solution with the addition of disodium glycerophosphate as a substance capable of absorbing hydrogen ions.

Despite many studies conducted on these systems [5,6,7,8,9,10,11,12,13,14,15,16,17], the mechanism of forming a three-dimensional polymer network is still under discussion, and its knowledge seems to be crucial in order to consciously design and manufacture in-situ gelling liquid systems.

The influence of the addition of glycerophosphate (GP) salt on the mechanism, sol-gel phase transition conditions and final mechanical properties of the obtained hydrogels remains a controversial issue. It is believed that this polyol salt can be responsible for electrostatic repulsion combined with the occurrence of ionic interactions [6], the formation of hydrophobic polymer-solvent interactions leading to the enhancement of polymer-polymer interactions [10] or the synergistic interaction of hydrogen bonds and hydrophobic interactions resulting in the formation of a gel [7]. Many studies have shown that the temperature of the sol-gel phase transition of chitosan systems is regulated by the glycerophosphate concentration. According to Filion et al. [8] and Lavertu et al. [9], the key role of the GP salt is its ability to thermostimulate absorption of protons released from chitosan; consequently, the formation of a gel network is related to the reduction of the dissociation degree of amino groups—reduce pKa by 0.023 units per K. As it results from the model proposed by Lavertu, the only proton recipient in the closed, limited outer cylinder with the radius R surrounding the chitosan polyion are GPO_4_^−^ (GP^−^) and GPO_4_^2−^ (GP^2−^) ions derived from glycerophosphate salt. The chitosan dissociation constant depends only on the concentration of monovalent and divalent glycerophosphate forms, and the type of cation associated with glycerophosphate does not affect the process mechanism. According to the authors, spatial confirmation of glycerophosphate salt is irrelevant, and it is even possible to use any phosphate salt, even inorganic.

A more interesting conclusion is the finding that glycerophosphate molecules are significantly responsible for the thermal sensitivity of polymer chains [11]. Initially, despite exceeding the chitosan isoelectric point of 6.3 [18], the addition of glycerophosphate salt prevents phase separation [19] and maintains the polymer in solution at physiological pH [11]. As a result, no macroscopic precipitation of chitosan from the solution is observed despite reaching a pH of about 7 [6]. According to Supper, this results from the ability of glycerophosphate molecules to adopt the appropriate spatial conformation, which protects the chitosan chains against precipitation from solution.

Applying the theoretical approach of colloid engineering [20,21], the authors in their previous research showed that the process of forming a spatial polymer network (gel) from a colloidal chitosan solution in the presence of GP salt proceeds in two stages. Since the amino groups present in the chitosan chain form non-covalent bonds with anionic phosphate groups, it was found that the aggregation rate of particles is not limited only by the phenomenon of their diffusion (diffusion-limited cluster aggregation (DLCA) mechanism [22,23,24,25]), but will also depend on the resulting interactions between the amino groups and anionic phosphate groups, so-called reaction-limited cluster aggregation (reaction-limited cluster aggregation (RLCA) mechanism [22,23,24,25]). In the first stage, the charge of the chitosan polyion is neutralized, leading to the precipitation of chitosan chains from the solution, thus creating aggregation nuclei. The kinetics of the process depends on the actual protonation state of the amino groups of the polyion. After the process of precipitation of chitosan chains, the actual gelation process takes place in the entire volume, the kinetics of which is regulated by the rate of nucleation formation as well as their diffusion; the process is limited by diffusion of polymer chains (DLCA mechanism). According to Owczarz et al. [20], the addition of disodium glycerophosphate leads to the occurrence of both aggregation mechanisms, clearly different in their progress rates. Moreover, based on the classic rheological measurements combined with the simultaneous small-angle light scattering analyses (SALS) [26], a significant influence of molecular weight on the spatial structure of chitosan in acid solution, temperature and kinetics of the phase transition, as well as differences in the activation energy values, was shown.

In the case of low molecular weight chitosan, its dissolution process related to the protonation of amino groups is difficult due to the conformation of the polysaccharide in the form of a rigid, impenetrable ball [20]. The longer the polymer chain is, the less entangled it is; the hydrogen ions can freely penetrate into the amino groups inside the structure. Due to the slow course of this aggregation regime, this stage must be considered first, and the most interesting concepts for describing the hydrogen ion transfer stage are the studies conducted by Filion and Lavertu cited above.

In order to increase the application potential of chitosan scaffolds, hybrid systems using silver nanoparticles [27] or zinc doped chitosan [28] are increasingly being produced. Simultaneously, an alternative to the commonly used sodium salt of glycerophosphate seems to be magnesium [29] and calcium [30,31] salts, which can improve the functional properties of the obtained scaffolds. These salts have so far been only an addition to the sodium salt, and the exclusive use of the calcium salt has been reported in the authors’ earlier studies. Owczarz et al. [32] showed that the use of the same concentrations of CaGP and NaGP leads to different conditions of the sol-gel phase transition, which contradicts the thesis given by Filion and Lavertu. Therefore, it seems reasonable to analyze the effect of glycerophosphate salts differing in cation on gelation conditions and to conduct a wider discussion in the context of the still discussed sol-gel phase transition mechanism of hybrid chitosan-glycerophosphate systems.

## 2. Results

### 2.1. pH and Zeta Potential Measurements

The pH values of the solutions obtained after dissolving chitosan in 0.1 M hydrochloric acid or acetic acid are 5.5 and 5, respectively. This means that when weak acetic acid is used, the protonation of the amino groups takes place much less than with hydrochloric acid. Figure 1 shows the effect of the GP salt concentration and the cation associated with it on the pH values of the tested systems. The addition of a buffering substance in the form of glycerophosphate led to an increase in the pH of the solution caused by the orthophosphoric acid properties. An increase in the concentration of glycerophosphate ions (GP^2−^) causes an increase in pH, depending on the type of cation associated with glycerophosphate. The stronger the base formed from the cation bound to glycerophosphate, the greater its influence on the increase of the pH of the colloidal chitosan system. Simultaneously, despite the exceedance of the isoelectric point, no macroscopic precipitation of the polymer was observed.

Figure 2 shows the results of changes in the Zeta potential value obtained during non-isothermal measurements. The analysis covered both the type of solvent, the effect of the origin and molecular weight of the chitosan, as well as the type of pH neutralizing substance, i.e., glycerophosphate, which differed in the type of attached cation. Regardless of the solvent used, it is evident that the measured Zeta potential values (ζ > 30) indicate high stability of the chitosan solutions; there is a stable double layer and the chitosan polyion chains are surrounded by water molecules; the system is hydrophilic.

The addition of glycerophosphate changes the counterions associated with the chitosan polyion. Chloride Cl^−^ and acetate CH_3_COO^−^ ions form salts with varying degrees of dissociation with sodium Na^+^, calcium Ca^2+^, and magnesium Mg^2+^ cations. Consequently, it causes a decrease in the value of the Zeta potential, which proves the destabilization of the system, the breakdown of the double layer, neutralization of the polyion chain, and the domination of its hydrophobic character. In the case of the addition of calcium glycerophosphate salt, it was found that in the range of 25–30 °C, depending on the chitosan and solvent used, a decrease in the value of the Zeta potential to the value of 0 was observed. This indicates the greatest instability of the system leading to particle aggregation.

The results of the Zeta potential measurements carried out at the sol-gel phase transition temperatures are shown in Figure 3. Compared to the results of measurements carried out under non-isothermal conditions, a much larger dispersion of the obtained points is visible; this phenomenon is especially observed for the sparingly soluble calcium salt and the magnesium salt with limited solubility strongly dependent on temperature (discussed below). Regardless of the repetitions made (*n* = 3–5), the course of the experimental data remained unchanged.

Similar to a study conducted under non-isothermal conditions, it was found that for all the glycerophosphates used, the colloidal chitosan systems show a very high instability. The greatest effect of the pH neutralizing substance was found in the case of disodium glycerophosphate salt, due to the high activity of sodium ion and the greatest increase in pH (Figure 1).

### 2.2. Perikinetic Aggregation

The rate of sol-gel phase transition of colloidal chitosan systems induced by stochastic Brownian motion was determined using turbidimetric techniques by measuring the values of transmittance T [%] and backscatter BS [%] signals in the entire sample volume [21]. Examples of experimental curves obtained in subsequent time intervals for a colloidal solution of chitosan acetate with the addition of disodium glycerophosphate are shown in Figure 4a.

Changes in the obtained transmittance and backscatter signals are presented as parallel lines with decreasing values of the measured quantity. This proves the change in the size of particles suspended in a continuous medium (more specifically about polymer aggregation). No apparent differences at the extreme ends of the height of the tested sample proves the lack of migration of the aggregation nuclei formed, only their merging into larger structures is observed, which leads to the production of an unlimited polymer lattice.

Figure 4b shows the curves recorded in subsequent time intervals for a colloidal solution of chitosan acetate with the addition of calcium glycerophosphate. In this case, the obtained transmittance T values are close to zero. This is due to the suspension of undissolved calcium glycerophosphate particles in the solution. When analyzing the backscatter curves and the increase in their value, it can be concluded that in subsequent intervals, a rapid aggregation of particles is observed, related to the increase in aggregate size. Completely different courses of transmittance and backscatter changes were obtained for systems containing magnesium glycerophosphate (Figure 4c). In this case, in the initial phase of the gelation process, the transmittance (T) increases with a simultaneous decrease in the backscatter signal (BS), which indicates that the magnesium glycerophosphate is dissolving and the solution is gradually clarifying. This may be due to an increase in the rate of the glycerophosphate dissociation reaction and an increase in the rate of hydrogen ion exchange between the amino groups of the chitosan polyion and the glycerophosphate. 

Figure 4d shows the course of the transmission curve obtained for the chitosan system with magnesium glycerophosphate, expressed as the difference between the current value and the initial value ΔT = T − Tp. The course of the transmittance curves clearly shows the stage of the process in which magnesium glycerophosphate dissolves (up to 2 h), followed by the actual chitosan aggregation process, visible as a decrease in the transmittance signal.

Since in all the considered cases, the destabilization of the solution occurs only in the form of particle aggregation and there are no sedimentation or creaming phenomena of suspended polymer aggregates, the kinetics of the process can be determined as changes in averaged transmittance or backscattering values for the entire height of the sample limited by the thickness of the boundary layers (2 mm from the bottom and top surface). Examples of the obtained curves for chitosan acetates with different molecular weights are shown in Figure 5.

In the case of chitosan systems with the addition of disodium glycerophosphate, the change of both signals is very rapid, and the shape of the curves obtained is similar to other systems characterized by lower critical solution temperature (LCST), e.g., cellulose derivatives [33]. A similar shape of the curves was also obtained for the magnesium salt. However, it is preceded by an increase in light transmittance, especially visible in the case of shrimp chitosan. This proves that the above-described MgGP salt dissolution is induced by temperature increase. The analysis of the results obtained for the systems with the addition of calcium glycerophosphate is a bit more complex to interpret. This is due to slight changes in the recorded values of both signals.

Although the determination of the phase transition conditions based on turbidimetric measurements is not as unambiguous as when using rheological measurements, the determination of the characteristic gelation time seems to be of key importance in the case of designing systems forming a spatial polymer network in heating chambers. In analogy to the analyses used for cellulose derivatives [34,35], the characteristic time was determined as the time in which the value of 50% of the relative transmittance was reached, obtained as a result of the normalization of the measurement data (Figure 6).

Based on the data obtained, it can be clearly indicated that the characteristic time TRel_50_ was achieved the fastest when using calcium salt (about 9 min). In the case of using the most commonly used sodium salt, the time was 40 min and in the case of magnesium glycerophosphate salt, the time was approximately 1500 min.

In order to take into account the total changes observed during the formation of the spatial structure, the Turbiscan Stability Index (TSI) parameter was determined, defining the total destabilization of the system, the changes of which can be successfully used to compare the course of the gelation process with the results obtained during rheometric measurements [21]. Based on the final TSI values (Figure 7a), it can be concluded that the greatest changes were observed in the case of the use of magnesium glycerophosphate, and the smallest in the case of the addition of calcium salt. In order to carry out a comparative analysis of the effect of glycerophosphate salts differing in cation, the values of the TSI parameter were normalized (Figure 7b).

The experimental curve obtained for the system containing the addition of disodium glycerophosphate takes the S-shape, which can be compared with the three characteristic areas distinguished during rheological measurements [10]. A fast gelation region, defined as a sharp change in the experimental parameter, is observed after about 100 min. In the case of the other two glycerophosphate salts, the experimental curves almost overlap up to 15 min. In the case of MgGP, a two-stage destabilization of the system was observed, which in the initial stage of the process corresponds to the hydrogen ion exchange stage between the chitosan polyion and glycerophosphate, leading to an increase in the solubility of this salt. Proper aggregation is observed in the second stage (after 400 min). In the case of calcium glycerophosphate salt, the fast gelation region lasts up to 30 min, then a slowdown in the growth of the polymer network is visible.

### 2.3. Rheological Study

#### 2.3.1. Mechanical Spectra

In the case of solutions containing equimolar amounts of glycerophosphate residues derived from the disodium, calcium, or magnesium salts, the obtained experimental curves of the storage modulus G′ and loss modulus G″ show the dependence of the viscoelastic properties of the medium on the salt-forming cation (Figure 8).

At a temperature of 5 °C, for low values of the angular frequency ω, the elastic properties dominate over the viscous ones (G′ > G″); the polymer is in a highly flexible state, plateau zone [36,37,38]. In this region, the thermal energy of the system exceeds the interaction energy of biomolecules. It is observed as sliding of the biomaterial chains in relation to each other. The increase in the angular frequency causes a change in the nature of the experimental media; there is a gradual domination of the viscous features over the elastic ones (G′ < G″)—the transition area to the glassy state is observed. Such behavior is exhibited by colloidal systems of chitosan chlorides (Figure 8a) and acetates (Figure 8f) with the addition of disodium or magnesium glycerophosphate. Regardless of the solvent used, in the case of the colloidal system with the addition of calcium glycerophosphate, the system shows high structure instability in the entire studied range of angular frequency ω.

In the case of systems containing disodium glycerophosphate and magnesium, the temperature increase causes only a reduction in the relaxation time, determined on the basis of the inverse of the angular frequency at which the curves of dynamic modules intersect. For samples with calcium glycerophosphate at a temperature of 25 °C, high system instability is visible; a phase transition process occurs in the colloidal chitosan system. Increasing the temperature to 30 °C terminates the fast gelation region. In the whole studied range of oscillation frequencies ω, the values of the storage modulus G′ are higher than the loss modulus G″ while simultaneously reaching the plateau; the polymer is in a glassy zone [36,37]. A further increase in temperature causes a gradual destabilization of the chitosan systems; first, the phase transition occurs when disodium glycerophosphate is used, and finally when a magnesium salt is added.

At 40 °C, colloidal chitosan solutions form an elastic network that is destroyed with slight deformation; the domination of elastic properties over viscous ones in the entire range of applied deformations is observed. In this area, mobility of individual monomers is observed with the simultaneous lack of movement of entire molecules. It should be noted that the elasticity of the formed network and its resistance to mechanical deformation depends on the type of cation associated with glycerophosphate salt. The storage modulus G′ reaches the highest values for the chitosan chloride system with the addition of disodium glycerophosphate; the obtained hydrogel network is characterized by the highest resistance to mechanical deformation.

In order to interpret the relationships between the viscoelastic functions, the mutual courses of the storage G′ and loss G″ moduli characterizing the current state of the intermolecular interaction in the solution, an analysis of the course of the dimensionless quantity, the damping factor tan(δ) defined as the ratio of the G″ to the G′, was performed (Figure 9).

The analysis of the obtained results showed that at low temperatures, regardless of the glycerophosphate salt used, the structure of a dilute crosslinked gel is formed [39]. The phenomenon of the presence of small crystallites and the penetration of solvent molecules inside the mesh is characteristic of these systems. When the sodium or magnesium salt is used, the increase in temperature leads to the formation of a weakly cross-linked amorphous polymer, the structure of which has the form of thread-like networks with very flexible chains (so-called soft rubber structure). On the contrary, in the case of systems with calcium glycerophosphate, it can be observed that at the highest temperature considered, a glassy structure is formed.

#### 2.3.2. Non-Isothermal Kinetic of Phase Transition

Figure 10 shows the changes in the values of the storage G′ and loss G″ moduli obtained during non-isothermal oscillatory measurements. Regardless of the tested solvent-GP salt system, in the initial heating phase the phenomenon of a decrease in the value of both dynamic modules, and thus the weakening of intermolecular interactions, is visible. This phenomenon and the increase in the intensity of thermal Brownian motion lead to an increase in the rate of hydrogen ion transfer, neutralization of amino groups and acceleration of the nucleation process and the subsequent stage of aggregation of the formed nuclei. It is observed as an area of rapid growth of the values of both modules; proper sol-gel phase transition. Finally, the expansion of the unlimited spatial network is observed, characterized by a very slow increase in the value of both dynamic modules.

Table 1 presents the values of the sol-gel phase transition temperature (gelation point) determined by the commonly used method of intersecting the curves of the modules tan(δ) = G″/G′ = 1 and the method based on the discontinuity of the function at the critical temperature point (the so-called spinodal decomposition temperature) proposed by Larson and Fredrickson [40]. On the basis of the determined phase transition point temperatures, it can be concluded that the method based on the equalization of dynamic modules only determines the beginning of the gelation process, i.e., the point at which the elastic properties of the tested medium begin to dominate over viscous properties. In the case of the Fredrickson–Larson (F-L) method, the determined gelation temperatures have a higher value and indicate the real point at which the aggregation mechanism changes from nucleation to the formation of a developed spatial structure. Only in the case of colloidal systems with calcium glycerophosphate are both temperatures similar. This results from the rapid kinetics of the phase transition.

Table 1 also shows the determined activation energy values of the fast gelation area [41,42,43] based on the analysis of changes in the storage modulus and the application of the polymer crystallization kinetics model based on the Arrhenius equation as well as the Arrhenius energy values in the viscoelastic flow region. It was found that the type of applied glycerophosphate significantly influenced the activation energy values in the fast gelation area, i.e., Region 1 and Region 2. As with the sodium glycerophosphate salt, the use of calcium and magnesium salts causes a two-stage stage of rapid development of the polymer network; the first is limited by reaction, the second by diffusion [20].

#### 2.3.3. Isothermal Kinetic of Phase Transition

Due to the practical application of the considered systems, special attention was paid to isothermal tests at 37 °C. Changes in the values of the storage G′ and the loss G″ moduli are shown in Figure 11. Regardless of the tested solvent—glycerophosphate system, similarly to the measurements carried out under non-isothermal conditions, a typical shape of experimental curves with three characteristic areas was distinguished. It has been found that in all the considered cases, the area of spatial structure formation observed as the range of rapid increase in the value of the modules was preceded by the occurrence of minimal values for both dynamic modules. It is extremely interesting that regardless of the solvent used as well as the glycerophosphate salt, the minimum value is achieved almost after the same measurement time (approximately 100 s for G′). In case of using calcium salt almost immediately after reaching the minimum value, the dynamic modules rocketed. The use of a magnesium salt results in a curve shape similar to that obtained for the most commonly used disodium glycerophosphate salt. This means that some delays in the occurrence of the fast gelation area are observed. This phenomenon directly affects the values of the determined gelation times (Table 2).

At the measurement temperature, the thermal energy of the system exceeds the energy of interaction of biomolecules (Figure 8). It is observed as the shifting of the biomaterial chains in relation to each other, weakening of mutual interactions, and thus a decrease in the complex viscosity of the fluid. It is especially visible in the case of a decrease in the storage modulus G′ value observed for chitosan acetate systems. Similarly to the measurements carried out under non-isothermal conditions, this indicates a weakening of intermolecular interactions and an increase in the intensity of thermal Brownian motion, which leads to an increase in the intensity of the hydrogen ion transfer reactions occurring in the system, neutralization of amino groups and acceleration of the nucleation phenomenon and subsequent aggregation of the embryos formed.

The data obtained during isothermal measurements were described using the Equation (1) proposed by the authors considering the change in structural properties occurring during the sol-gel phase transition in the fast gelation region. In the considerations, the Avrami nucleation theory [44] and the basics of crystallization described by Ziabicki [45,46,47] were applied. The equation takes into account the state of the structure of the colloidal polymer system after the precipitation of polymer chains, formation of unstable embryos, and aggregation and formation of aggregation nuclei with a dimension greater than the critical ones v * [48].
G′(t) − G′(t_0_) = A∙R∙T∙exp[k∙(t − t_0_)](1)
A = (v_2_^(1/3)^)/M_c_(2)

Table 2 shows the determined values of parameters A and k. Much higher values of the parameter k confirm quantitatively that in the case of solutions containing calcium glycerophosphate, the structure changes occur very rapidly. At the same time, according to the Equation (2), the low value of the A coefficient means that even before the area of rapid growth of the structure, the occurrence of aggregates with a high value of the number average molecular weight of the polymer chains between adjacent crosslinks is observed.

### 2.4. FTIR Spectroscopy

Figure 12 shows the Fourier Transform Infrared Spectroscopy (FTIR) spectra obtained for lyophilized chitosan hydrogels obtained from solutions of chitosan chlorides and acetates containing disodium, calcium, as well as magnesium glycerophosphate.

In the spectra obtained, the presence of peaks in the wavenumber range 3600–2500 cm^−1^ and 1700–500 cm^−1^ were found. The similarity of the position of the sample peaks, containing respectively disodium, calcium or magnesium glycerophosphate, can be seen in the wavenumber range 3600–3000 cm^−1^. The peak obtained for the wavenumber 3293–3273 cm^−1^ indicates the presence of stretching vibrations of the N-H groups. For all analyzed colloidal systems, the occurrence of three repeating peaks was observed for the following wavenumbers: 2159 cm^−1^, 2023 cm^−1^ or 2019 cm^−1^, and 1976 cm^−1^.

In the case of using calcium glycerophosphate and acetic acid as a solvent, an increase in the relative number of bonds characteristic for peaks in the 1750–1400 cm^−1^ wavenumber area is visible, compared to the bands in the 1200–900 cm^−1^ range. The presence of peaks for the 1080 cm^−1^ and 997 cm^−1^ wavenumbers were also found, which indicate the presence of stretching vibrations of the C-O group and bending vibrations of the C = C bond. The presence of these two peaks in an asymmetric, convergent form is characteristic of the presence of the -PO_4_^3−^ group. For the wavenumber range 1200–900 cm^−1^, the presence of one intense peak was found for 1055 cm^−1^. The occurrence of one peak with a similar shape for slightly higher values of the wavenumber was described in the case of scaffolds containing disodium glycerophosphate. Below the wavenumber of 1000 cm^−1^, the area characteristic for the presence of -HPO_4_^2−^ groups is observed. This region is characterized by a different course and intensity depending on the glycerophosphate used. The most significant of the observed differences in the absorbance intensity, resulting from the incomplete silencing of the peaks for the 2550 cm^−1^ and 2159 cm^−1^ wavenumbers. This may indicate the interaction between the stretching vibrations of the O-H group in the carboxylic acid.

The addition of a glycerophosphate salt other than sodium into the system causes a decrease in the intensity of the peaks for the wavenumber 3293 cm^−1^ and 2548 cm^−1^ and the appearance of a new peak for the wavenumber 1125 cm^−1^. The latter indicates the presence of stretching vibrations of the C-O group.

The main difference observed in the FTIR spectra obtained from chitosan chlorides and acetates is the presence of peaks in the wavenumber range 2800–1800 cm^−1^, e.g., the presence of a new band for 2550 cm^−1^. As mentioned above, the presence of this band indicates the presence of stretching vibrations of the O-H groups in the carboxylic acid. However, the most significant differences resulting from the use of different solvents are visible in the wavenumber range of 1300–1600 cm^−1^. The vibrations occurring in this range correspond to the C-H bonds of the acetic acid alkane chain and do not occur in the case of using hydrochloric acid as a solvent.

## 3. Discussion

The complexity of the chitosan gelation process currently prevents an unambiguous description of the phenomenon, while the proposed mechanisms are very often single-threaded and postulated on the basis of indirect evidence. In the literature, there is no analysis of the effect of the solvent type and the unambiguous role of glycerophosphate in the chitosan cross-linking process. It is known that the use of appropriate concentrations of glycerophosphate leads to the sol-gel phase transition point at a temperature of approximately 37 °C [5]; additionally, depending on the pH value, chitosan molecules assume a different spatial conformation from a statistical Gaussian coil to an extended, rod-like conformation. The latter form of the polymer promotes the formation of non-covalent bonds between adjacent polymer chains and leads to their aggregation, while the compact, coiled spatial conformation of the polymer chain affects the degree of protonation of the amino groups, hindering the diffusion of hydrogen ions into the Gaussian coil [26].

### 3.1. Influence of the Solvent Type

Chitosan as a cationic polymer is soluble in aqueous solutions whose pH value is lower than 6.2. Proposed by Chenite et al. [5,6], the solution of obtaining colloidal chitosan systems is based on dissolving chitosan in 0.1 mol/dm^3^ hydrochloric acid solution. However, it is known that it is also possible to use a solution of another monobasic acid (e.g., formic acid, acetic acid, propionic acid, or lactic acid). Since the most commonly used organic solvent is 0.1 mol/dm^3^ acetic acid solution, this acid will be compared with hydrochloric acid [5,6,10,20,37,49,50].

During the dissolution of chitosan, the dissociation reactions of the respective acids take place and, at the same time, the protonation reaction of the amino groups in the chitosan chain, which is given by Equation (3):(n∙m) H_3_O^+^ + m∙CS (n∙NH_2_) ↔ m∙CS (n∙NH_3_^+^) + (n∙m) H_2_O(3)
where m is the molarity of chitosan and n isthe number of ammonium groups in a single polymer molecule (corresponds to the deacetylation degree (DD) of chitosan).

If we assume that the molar concentration of the amino groups associated with the chitosan chain is given by Equation (4):(4)$$c_{\mathrm{CS}\left({\mathrm{NH}}_{2}\right)\;}={\;c}_{\mathrm{CS}}\cdot \mathrm{DD}$$
then Equation (3) will take the form according to Equation (5):(5)$$H_{3}O^{+}+\mathrm{CS}{\left({\mathrm{NH}}_{2}\right)}^{}↔\;\mathrm{CS}{\left({\mathrm{NH}}_{3}^{+}\right)}^{}+H_{2}O$$

Due to the different strength of the solvents, it should be expected that the degree of protonation of the amino groups, and thus the final pH of the solutions of chitosan chlorides and acetates, will be different. In the case of using hydrochloric acid (dissociation constant Ka__HCl_ = 10^7^, pKa_HCl_ = −7), a solution of a salt of a strong acid and a weak base (a derivative of amino groups) is obtained. In this case, the pH of the chitosan chloride solution will depend only on the concentration of the protonated amino groups $c_{\mathrm{CS}\left({\mathrm{NH}}_{3}^{+}\right)}$. The value of the hydrolysis constant of chitosan chloride can be calculated from Equation (6) and the pH can be calculated from Equation (7).
(6)$$K_{h}=\frac{K_{w}}{K_{\mathrm{BCS}\left({\mathrm{NH}}_{2}^{}\right)}}\;$$
where K_w_ is water dissociation constant.
(7)$$\mathrm{pH}=\;-\log\left[H_{3}O^{+}\right]\;\mathrm{where}:\left[H_{3}O^{+}\right]=\;\sqrt{\frac{K_{w}\cdot c_{\mathrm{CS}\left({\mathrm{NH}}_{3}^{+}\right)}}{K_{\mathrm{BCS}\left({\mathrm{NH}}_{2}^{}\right)}}}$$

When using acetic acid as a solvent (dissociation constant K_A___CH3COOH_. = 1.8 × 10^−5^, pKa_CH3COOH_ = 4.76), a salt of a weak acid and a weak base will be obtained. In this case, the hydrolysis constant is given by the Equation (8):(8)$$K_{h}=\frac{K_{w}}{K_{A\_\mathrm{CH}3\mathrm{COOH}}\cdot K_{\mathrm{BCS}\left({\mathrm{NH}}_{2}\right)}}\;$$
and the pH value can be calculated from the Formula (9):(9)$$\mathrm{pH}=\;-\log\left[H_{3}O^{+}\right]\;\mathrm{where}:\left[H_{3}O^{+}\right]=\;\sqrt{\frac{K_{w}\cdot K_{A\_\mathrm{CH}3\mathrm{COOH}}}{K_{\mathrm{BCS}\left({\mathrm{NH}}_{2}^{}\right)}}}$$

In the case of simultaneous presence of ammonium and acetate ions, we deal with a special system when the pH of the solution of the resulting salt in perfectly pure water will be equal to 7 because the acid has the same dissociation constant value as the base associated with it in the solution. It should be noted that such a value can be achieved only in the case of complete protonation of the ammonium groups in the chitosan chains.

Based on the conducted research, it was found that in the case of weak acetic acid, the protonation of the amino groups occurs to a much lesser degree than in the case of hydrochloric acid. The pH values measured after 24 h of storage at room temperature were pH = 5 for acetic acid and pH = 5.5 for hydrochloric acid, respectively (Figure 1). The results obtained with acetic acid as the solvent are of particular interest. As described above, the pH value is expected to equal 7. The lower value is due to the less than 100% degree of protonation of the amino groups and the hydrophilic-hydrophobic interactions of the chitosan polyion with the solvent particles, which lead to the adoption of different spatial conformations of the chitosan chains, from the statistical Gaussian coil to the extended rod-like conformation. The compact, folded conformation of the polymer chain affects the degree of protonation of the amino groups, hindering the diffusion of hydrogen ions into the Gaussian coil [20]. In view of the above, the assumption of 100% protonation of the amino groups is too far-reaching, and the conformation of the polymer chains in the solution prevents their complete ionization.

### 3.2. pH Neutralization; Glycerophophate Salt-Solvent Interaction

The addition of glycerophosphate to the chitosan salt solution causes destabilization of the system and a decrease in the value of the Zeta potential, which proves the breakdown of the double layer, neutralization of the polyion chain, and its hydrophobic character dominating. Consequently, this leads to the precipitation of the polymer from solution and aggregation of the particles leading to the formation of the lattice. However, the analysis of the impact of individual solution parameters is not unequivocal. Despite the fact that the concentrations of the amino groups for all the chitosans used were the same (taking into account the weight, monomer weight, and the degree of DD deacetylation), the influence of temperature on colloidal systems with different chain lengths was different. Only in the case of solutions containing calcium glycerophosphate in both solvents used did the destabilization take place in the same order, not resulting from the weight average molecular weights of the polymer. The greatest influence of temperature is visible in the case of crab chitosan, which is also most often described in the literature.

The lack of unambiguity of the results of the Zeta potential measurements (Figure 2) may also indicate a different spatial conformation of the polymer chains depending on the molecular weight of the polymer, the solvent used and the cation associated with glycerophosphate.

The addition of glycerophosphate changes the counterions associated with the chitosan polyion. Chloride and acetate ions form salts with varying degrees of dissociation with sodium, calcium, and magnesium cations; the role of the counterion is taken over by glycerophosphate ions. In solutions with a high content of inert salts, the charges of the polyion are screened and the polyelectrolyte chain takes on a random coil conformation, characteristic of non-ionic polymers.

Moreover, considering the applied electrophoresis method as a method of measuring the Zeta potential, it can be assumed that the measured value depends not only on the current thickness of the double layer but also on the conformation of the polymer chain. The shape and packing of the polymer affect the flow resistance of particles in the electric field. Whereas, according to Henry’s equation, electrophoretic mobility is the basis for determining the value of the Zeta potential.

### 3.3. Influence of the Cation Type of on the System Destabilization and the Phase Transition Kinetics

The most frequently used method of lowering the phase transition temperature of chitosan is the use of a polyol, i.e., a salt of β-disodium glycerophosphate [5,6], which simultaneously increases the pH above 6.2. Phosphoric acid is a polybasic acid and has the ability to gradually lose or attach hydrogen ions. Considering the buffer properties of the strong base phosphate salt first, the form of the ion formed as a result of dissociation depends on the current pH value in the system. Consequently, ions with varying degrees of hydrolysis to both GP^2−^ and HGP^−^ are present in the solution.

Taking into account the two-stage dissociation of disodium glycerophosphate, the equations of the proton exchange process between amino groups and disodium monoglycerophosphate molecules in an acidic solution can be written as shown in the Equations (10) and (11):(10)$${\mathrm{Na}}_{2}{\mathrm{GPO}}_{4}+H_{2}O↔{\;\mathrm{HO}}^{-}+{\mathrm{NaHGPO}}_{4}^{}+{\mathrm{Na}}^{+}$$
(11)$${\mathrm{NaHGPO}}_{4}^{}+H_{2}O↔{\;\mathrm{HO}}^{-}+H_{2}{\mathrm{GPO}}_{4}^{}+{\mathrm{Na}}^{+}$$
where G is a glycerin molecule to which an orthophosphoric acid residue (V) is attached.

DuBois [51] points out that despite the same chemical composition, the attachment place of the orthophosphate (V) group has a significant impact on the properties of disodium monoglycerophosphate. The most important difference is the solubility both in water and in acidic solvents; the α-glycophosphate form being much more soluble than the β-isomer. At the same time, the author suggests that the presence of organic acids (acetic acid, citric acid, etc.) increases the solubility of both isomers of monoglycerophosphate.

When chitosan hydrogels are used as scaffolds for cell culture, magnesium or calcium would be more preferred phosphorus related elements. Since both elements form strong bases, it should be assumed that the buffering solutions will be obtained and, as in the case of using sodium compounds (the same values of the hydrolysis constants), the effect of neutralizing the charge of the amino groups will be achieved. However, both calcium and magnesium phosphate are sparingly soluble salts; their solubility decreases with increasing cation atomic weight, and increases with the number of hydrogen atoms in the molecule of the orthophosphate salt, the dihydrogen orthophosphate (V) salts are more soluble than the monohydrogen ones.

The studies of the gelation of chitosan systems in the presence of different cationic glycerophosphate salts performed using the turbidimetric technique confirm the influence of the solubility of these salts on the phase transition kinetics under isothermal conditions. Changes in transmittance and backscatter values indicate a change in the size of particles suspended in a continuous medium, i.e., polymer aggregation. No apparent differences at the extreme ends of the height of the tested sample proves the lack of migration of the aggregation nuclei formed, only their merging into larger structures is observed, which leads to the production of an unlimited polymer network. Distinct differences in the values of the parameters determined indicate the presence of undissolved calcium glycerophosphate salts in the system (transmittance values close to zero) compared to an almost transparent solution containing NaGP. In the case of using MgGP salt, the progressive solubility of this compound is visible, until the destabilization of the chitosan colloid and the subsequent formation of the hydrogel structure (Figure 6c and Figure 7b).

The results of rheological measurements confirmed the effect of the cation bound with the glycerophosphate salt on the location of the gelation point—under non-isothermal (Table 1) and isothermal (Table 2) conditions. The use of the same concentration of GP^2−^ glycerophosphate ions associated with magnesium ion, as in the case of the most commonly used disodium glycerophosphate, causes a slight increase in the phase transition temperature (Table 1) while accelerating the gelation process under isothermal conditions (Table 2). Completely different results were obtained when using calcium glycerophosphate as a pH buffer. In this case, both the time of the phase transition and its temperature reached much lower values. When the same concentrations of glycerophosphate ions were used, the phase transition temperature dropped from about 39 °C for sodium salt to about 20 °C for calcium ones (Table 1). A slightly different shape of the storage and loss moduli curves was also observed, which could indicate differences in the phase transition mechanism. This may be due to the poor solubility of calcium glycerophosphate—confirmed by turbidimetric measurements. The comparison of the determined Arrhenius energy values obtained for the investigated systems shows that when CaGP salts are used, the gelation process in Region 1 and 2 is associated with the highest accumulation of supplied energy. Simultaneously, this process is the fastest in the regions concerned. Undissolved crystals of this salt are the natural aggregation nuclei of chitosan chains.

Thus, it was shown that the type of cation associated with the glycerophosphate salt as well as its solubility influences the gelation process of chitosan colloids much greater than the type of solvent used. Based on the conducted analyses, it was shown that ensuring the same proportions of glycerophosphate ions differing in cation to amino groups present in chitosan chains leads to obtaining systems significantly different in viscoelastic properties and phase transition conditions. Therefore, the assumption made by Lavertu and Filion that only the presence of an appropriate concentration of phosphate groups is a sufficient stimulus to carry out the gelation process is not correct.

Figure 13 shows a schematic course of the normalized curves of absorbance, destabilization coefficient, as well as rheological parameters describing the change in the structural properties of the medium, such as the storage modulus G′ and complex viscosity η*. According to the presented concept, the gelation process consists of a phase transition from chitosan sol (liquid-like phase) to a hydrogel form, a porous solid phase resulting from phase inversion and the entrapment of the solvent inside the lattice.

The pattern curves characteristic for a given measurement method, corresponding to actual changes in experimental parameters, indicate the presence of three characteristic regions. The first diffusion stage (for turbidimetric measurements in the perikinetic regime) or the viscoelastic flow stage (for rheometric measurements (orthokinetic)) ends with the formation of thermodynamically stable aggregation (crystallization) nuclei. The nuclei formed in this stage undergo further rapid aggregation in the second stage, the so-called fast gelation region. The differences in the course of the curves visible in the first stage result from the process regime, perikinetic or orthokinetic [21].

Due to the new information and the so far unused approach in the case of colloidal chitosan systems, we believe that in order to formulate an unambiguous mechanism, further studies should be carried out to thoroughly investigate the chemical phenomena occurring during the reaction limited aggregation, the stage of neutralization of polyion chains, and the formation of aggregation nuclei. Simultaneously, the second direction of further research is the precise determination of the role of glycerophosphate ions or glycerol resulting from their hydrolysis in the process of forming the polymer lattice. This is due to the fact that the presence of glycerin molecules in the system may lead to the formation of non-covalent bonds between the hydroxyl groups (-OH) and the amino groups (-NH_2_) of the chitosan chain.

## 4. Materials and Methods

The studies used chitosan of various origins and different molecular weights: chitosan of crab (Sigma Aldrich, product No. 50494), shrimp (Sigma Aldrich, Poznan, Poland, product No. 50494), as well as chitosan from the Fluka company of unknown origin (Fluka Analitycal provided by Alchem, Torun, Poland, product No. 50949) [20]; 0.1 M solutions of hydrochloric acid (Fluka Analitycal provided by Alchem, Torun, Poland, product number: 84415) and acetic acid (SigmaAldrich, Poznan, Poland, product No. 695092) were used as monobasic solvents. Disodium glycerophosphate, calcium glycerophosphate, and magnesium glycerophosphate (Table 3) were used as a substance capable of absorbing hydrogen ions and neutralizing the amino groups of the chitosan polyion.

Four hundred mg of chitosan powder was dissolved in 16 g of hydrochloric acid or 20 g of acetic acid, respectively. The resulting chitosan solution was left for 24 h to completely dissolve the polysaccharide. At this time, the solution was cooled to 4 °C over 2 h. Simultaneously, a suspension of glycerophosphate salt was prepared by introducing 2 mL of distilled water to a strictly defined amount of powder (Table 3). The suspension, cooled for 2 h, was added drop by drop to the colloidal chitosan solution. After thorough mixing, the samples were stored for 24 h at 4 °C to completely eliminate air bubbles. During the preparation of the experimental material, the commonly used methodology was followed [17,36].

### 4.1. pH and Zeta Potential Measurements

The pH measurements were performed at 15 °C in a thermostatic bath using the ELMETRON CP-401 pH-meter (ELMETRON Sp. J., Zabrze, Poland), equipped with a glass electrode for viscous liquids ERH-12-6.

The Zeta potential was measured by analyzing the electrophoretic mobility of the molecule in an electric field using the ZetaSizer Nano ZS device (Malvern Instruments Ltd., Malvern, UK). A capillary measuring cell of the DTS1060 type was used. Measurements were carried out under both isothermal and non-isothermal conditions; all tests were repeated 3 to 5 times.

In the case of isothermal measurements, the measuring cell was filled with the medium at a storage temperature of 5 °C and placed in the measuring chamber of the device. The tests were carried out at a constant temperature, close to the phase transition temperature, which was 40 °C for chitosan solutions containing disodium or magnesium glycerophosphate, and 20 °C for solutions containing calcium glycerophosphate, respectively.

The tests under non-isothermal conditions were carried out in the temperature range from 5 °C to 50 °C. The temperature change interval was 2 °C, and the heating rate was 1 deg/min.

### 4.2. Turbidimetric Study

Analyses were performed using a Turbiscan Lab turbidimeter (Formulaction, Toulouse, France) with standard 20 mL measuring cells. The measurements were carried out under isothermal conditions at 38 °C ± 0.5 °C. The measuring cell was filled with the medium at the storage temperature of 5 °C and placed in the measuring system.

### 4.3. Rheological Study

Measurements of rheological properties were carried out using an Anton Paar Physica MCR301 rotational rheometer (Anton Paar, Warszawa, Poland) using a cone-plate measuring system of 50 mm diameter, 1° cone angle, and 0.048 mm cone truncation.

The sol-gel phase transition conditions and the dynamics of the gelation process were examined during measurements under isothermal (37 °C) and non-isothermal conditions (from 5 °C to 60 °C, heating rate 1 deg/min) with constant deformation (angular frequency ω = 5 rad/s and amplitude γ = 1%).

Mechanical properties were determined in isothermal angular frequency sweep tests, with a constant amplitude equal to 1%, the value of which was determined in the linear viscoelasticity tests. The tests were performed at 5 °C, 25 °C, 30 °C, 35 °C, and 40 °C.

### 4.4. Fourier Transform Infrared Spectroscopy (FTIR) Study

FTIR spectra of freeze-dried chitosan gels were obtained with Nicolet™ iS™10 FT-IR apparatus equipped with a monolithic diamond ATR crystal (Thermo Scientific Inc., Waltham, MA, USA). The measurements were carried out in the range 4000–500 cm^−1^, with a resolution of 4 cm^−1^.

## 5. Patents

The research presented in the manuscript has been patented at Patent Office of the Republic of Poland (Pat.235629).

## Figures and Tables

**Figure 1 ijms-22-04043-f001:**
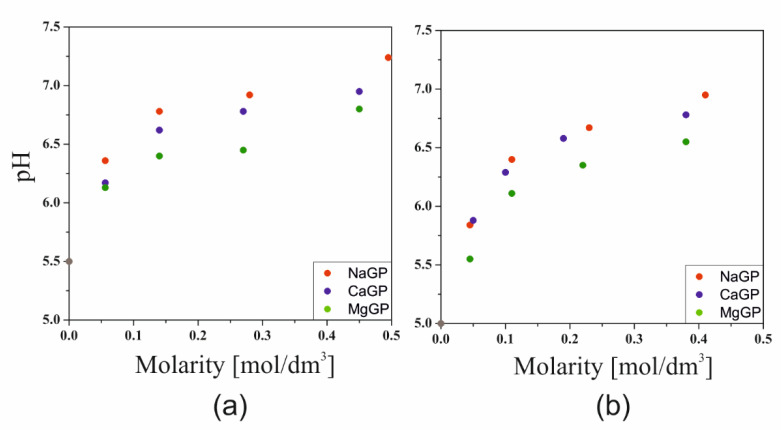
Changes in the pH value of solutions of crab chitosan chlorides (**a**) and acetates (**b**) as a function of glycerophosphate concentration. The data for the system containing the addition of calcium glycerophosphate using acetic acid come from the authors’ own research published earlier [32].

**Figure 2 ijms-22-04043-f002:**
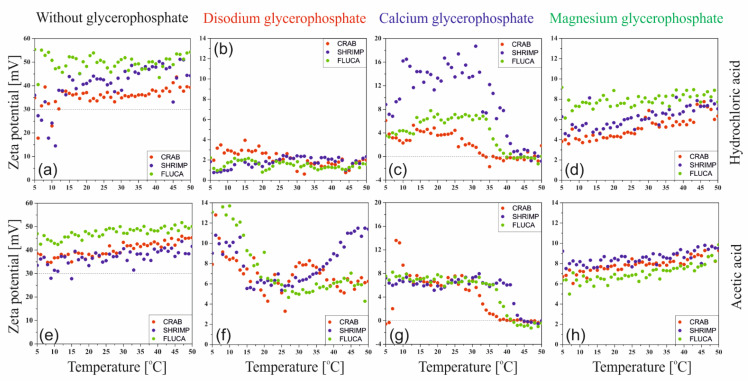
Changes in the Zeta potential value determined during heating for chitosan chlorides (**a**–**d**) and acetates (**e**–**h**) without and with the addition of various glycerophosphate salts.

**Figure 3 ijms-22-04043-f003:**
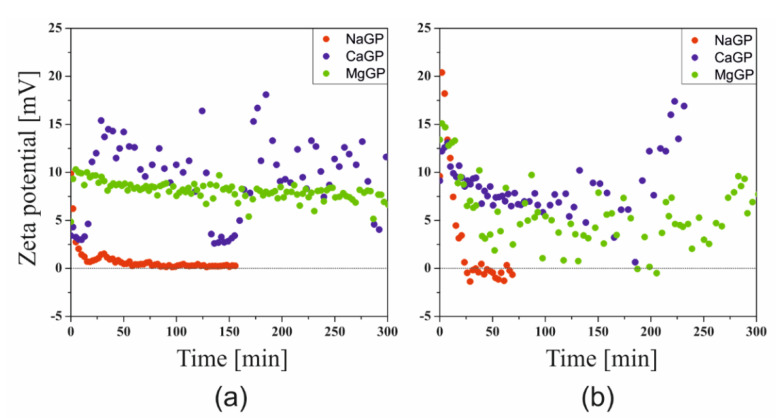
Kinetics of changes in the value of the Zeta potential of solutions of chlorides (**a**) and acetates (**b**) of crab chitosan.

**Figure 4 ijms-22-04043-f004:**
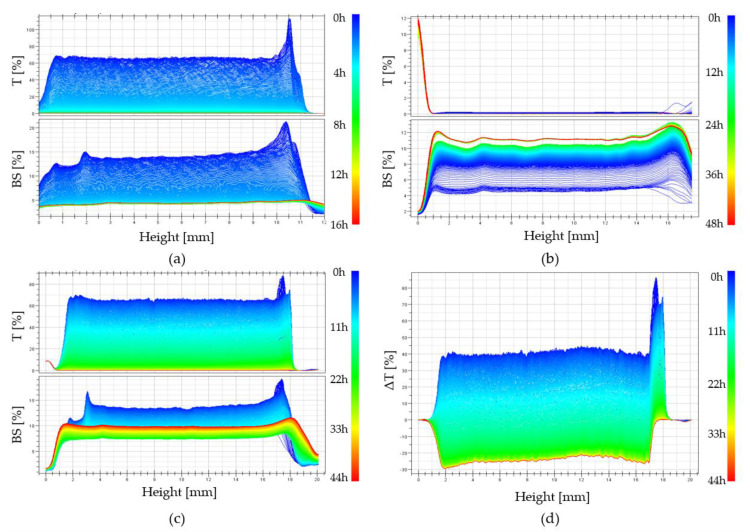
The intensity curves of the transmittance (T) [%] and the backscattering (BS) [%] reflected from the suspended particles obtained for the colloidal systems of chitosan acetate containing (**a**) disodium glycerophosphate, (**b**) calcium glycerophosphate, and (**c**) magnesium glycerophosphate. (**d**) Differential transmittance signal curves ΔT = T − Tp [%] obtained for a colloidal system of chitosan acetate containing magnesium glycerophosphate.

**Figure 5 ijms-22-04043-f005:**
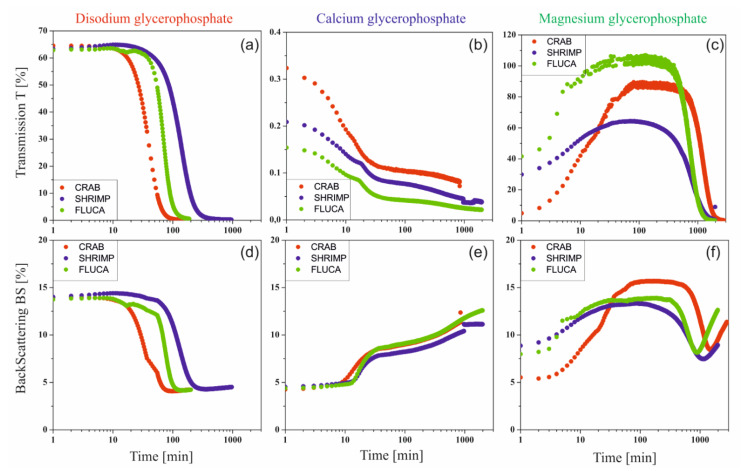
Curves of averaged transmittance (**a**–**c**) and backscatter (**d**–**f**) values obtained for a colloidal system of chitosan acetates with different molecular weights containing disodium, calcium or magnesium glycerophosphate salt.

**Figure 6 ijms-22-04043-f006:**
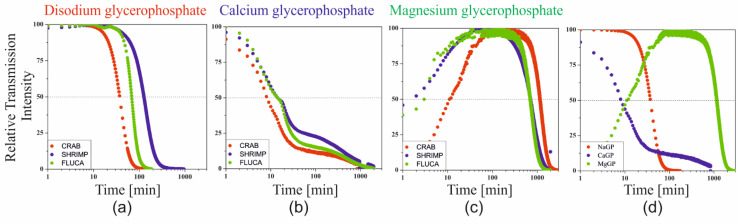
Changes in the relative intensity of transmittance for chitosan systems containing (**a**) sodium, (**b**) calcium, and (**c**) magnesium glycerophosphate. (**d**) Comparison of the relative transmission for the system obtained from crab-derived chitosan.

**Figure 7 ijms-22-04043-f007:**
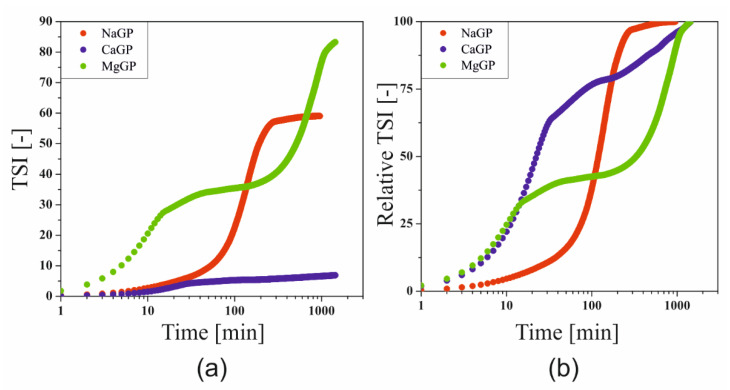
Influence of the type of glycerophosphate salt on the change in the value of the (**a**) destabilization and (**b**) relative destabilization parameters.

**Figure 8 ijms-22-04043-f008:**
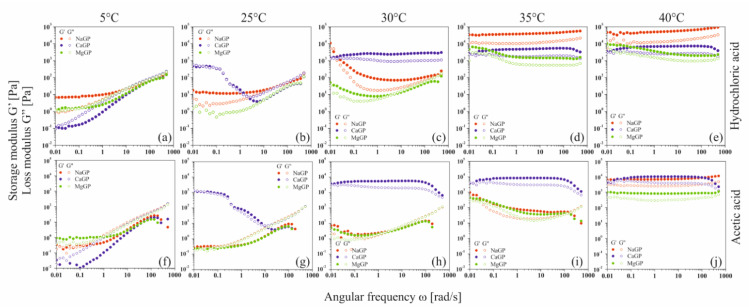
Mechanical spectra of chitosan chlorides (**a**–**e**) and acetates (**f**–**j**) containing disodium, calcium, or magnesium glycerophosphate, respectively.

**Figure 9 ijms-22-04043-f009:**
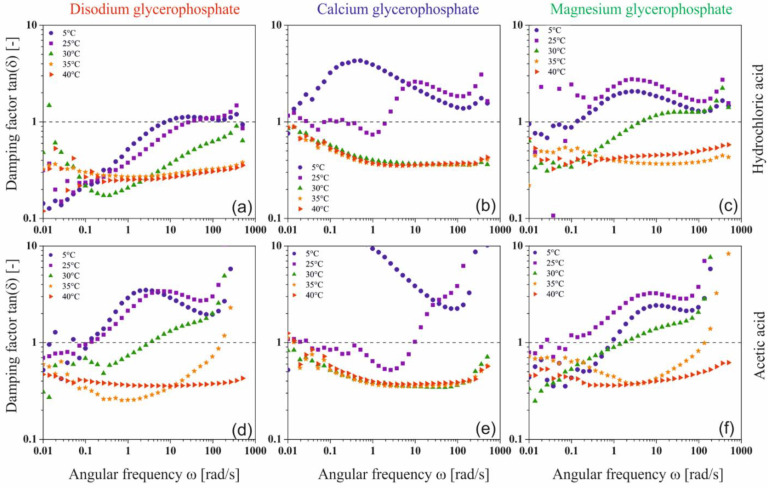
Changes in the value of the damping factor for chitosan chloride (**a**–**c**) and acetate (**d**–**f**) systems containing sodium glycerophosphate, calcium glycerophosphate, and magnesium glycerophosphate.

**Figure 10 ijms-22-04043-f010:**
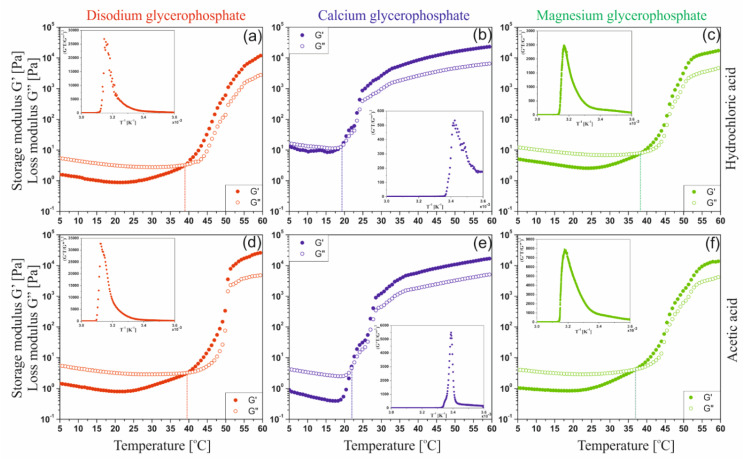
Experimental curves of changes in the values of the storage G′ and loss G″ moduli obtained during non-isothermal measurements for chloride (**a**–**c**) and acetate (**d**–**f**) chitosan solutions containing sodium glycerophosphate, calcium glycerophosphate, and magnesium glycerophosphate.

**Figure 11 ijms-22-04043-f011:**
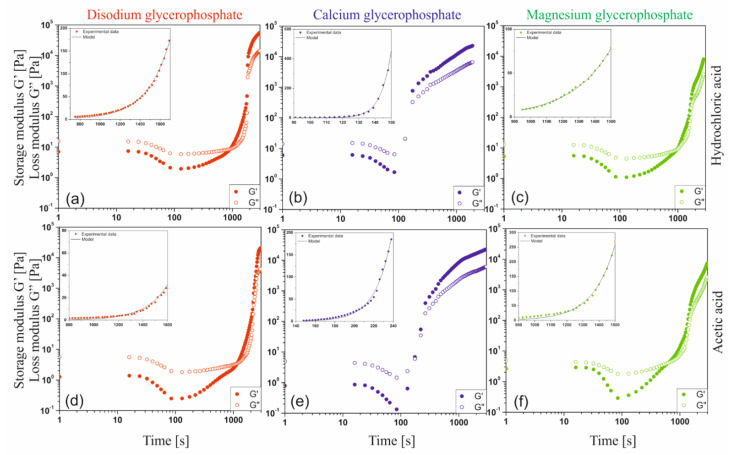
Experimental curves of changes in the values of the storage G′ and loss G″ moduli obtained during isothermal measurements at 37 °C for chloride (**a**–**c**) and acetate (**d**–**f**) chitosan solutions containing sodium glycerophosphate, calcium glycerophosphate, and magnesium glycerophosphate.

**Figure 12 ijms-22-04043-f012:**
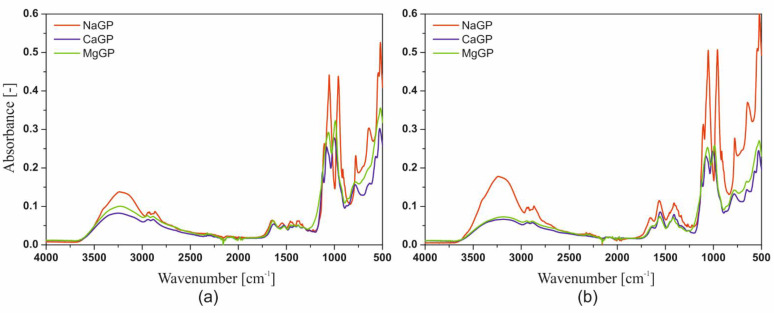
Fourier Transform Infrared Spectroscopy (FTIR) spectra obtained for chitosan chlorides (**a**) and acetates (**b**) with the addition of disodium glycerophosphate (red), calcium glycerophosphate (blue), or magnesium glycerophosphate (green).

**Figure 13 ijms-22-04043-f013:**
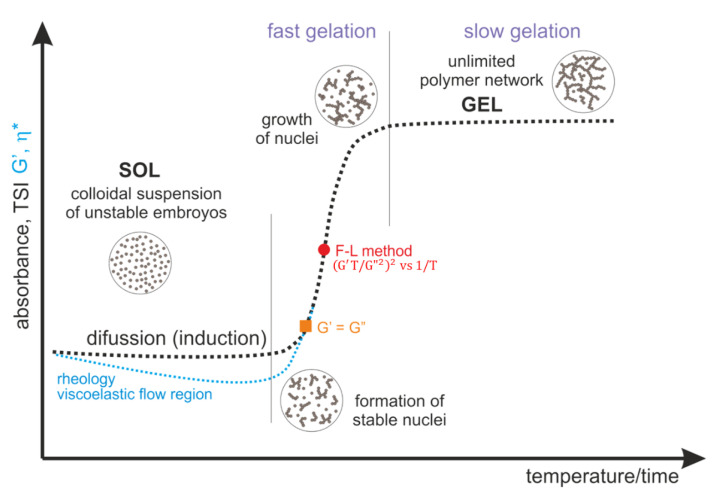
Characteristic parameters course during the sol-gel phase transition of colloidal chitosan systems, taking into account the formation of stable nuclei and their further aggregation.

**Table 1 ijms-22-04043-t001:** Arrhenius energy values in the fast gelation area and energy in the viscoelastic flow region as well as the phase transition temperature.

Glycerophosphate (GP) Salt	Solvent	Arrhenius Energy Ea (kJ.mol^−1^)			Gelation Temperature T gel (°C)	
		Viscoelastic Flow Region.	Region 1	Region 2	tan(δ) = 1	F-L Method
NaGP	HCl	24.6	288.7 ±31.3	1978.0 ±636	38.7	45.9
	CH_3_COOH	25.0	506.6 ±25.5	506.6 ±25.5	39.6	43.0
CaGP	HCl	21.5	695.2	970.4	19.0	19.3
	CH_3_COOH	28.6	795.7 ±44.5	1074.0 ±281	21.7	22.1
MgGP	HCl	22.3	141.3 ±21.2	631.9 ±31.8	38.4	42.4
	CH_3_COOH	17.6	64.2	557.0	36.9	41.5

**Table 2 ijms-22-04043-t002:** Values of parameters A and k of Equation (1) and the gelation time.

GP Salt	Solvent	$A=\frac{v_{2}^{\frac{1}{3}}}{M_{c}}$	*k*	R^2^	Gelation Time
		[Pa]	[s^−1^]	[-]	[s]
NaGP	HCl	0.1081	0.00435	0.9959	916
	CH_3_COOH	0.004335	0.00556	0.9879	1160
CaGP	HCl	2.70 × 10^−8^	0.1569	0.9955	130
	CH_3_COOH	0.0001186	0.05998	0.9942	166
MgGP	HCl	0.1823	0.00404	0.9984	958
	CH_3_COOH	0.004071	0.00737	0.9915	631

**Table 3 ijms-22-04043-t003:** Summary of glycerophosphate salts used in research.

	Disodium Glycerophosphate	Calcium Glycerophosphate	Magnesium Glycerophosphate
Manufacturer	Sigma Aldrich Poznan, Poland	Sigma Aldrich Poznan, Poland	Sigma Aldrich Poznan, Poland
Serial number	50020	G6626	17766
CAS number	13408-09-8	58409-70-4	927-20-8
Molecular weight (g/mol)	306.11	210.14	194.36
Conformation	β	β	α
Salt form	salt pentahydrate	anhydride	anhydride
Powder mass (g)	2.00	1.37	1.27
Mark	NaGP	CaGP	MgGP

## Data Availability

The data presented in this study are available on request from the corresponding author.

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
