# Peer review of "Influence of Glycerophosphate Salt Solubility on the Gelation Mechanism of Colloidal Chitosan Systems"

_ijms, 2021, doi:10.3390/ijms22084043_

Round 1

Reviewer 1 Report

This is a very interesting paper that could be published almost as-received since it is based on some sound experimental observations which show the influence of the cation used together with the glycerphosphate anion on the  stability/gelation process for various types of chitosan solutions in the presence of either acetic acid or HCl. However, I do have some questions that the authors need to address, given below:

Figure 1. Why is there scatter on the concentration axis?

Figure 3. It looks like the data is highly variable and hence questionable. How many repeats were performed? Can the profiles shown be accurately reproduced every time? I realize that such things are not uncommon in these measurements and generally speaking is indicative of instability, but the authors need to insert some text by way of specific explanation here.

Author Response

We are grateful for the review of our manuscript and its positive evaluation.  Please find the attachment with detailed answers and explanations.

Reviewer 2 Report

Present manuscript by Owczarz et al. entitled “Influence of glycerophosphate salt solubility on the gelation mechanism of colloidal chitosan systems” Overall the structure of this study is good and of benefit to larger community and industry. I recommend minor revision for this manuscript before it could be published in the IJMS.

Comments:

  1. Author should correct spelling missing/error in the manuscript like elastic ones (G "<G"); line number 266.
  2. Reference should be added in introduction: Cellulose, 26, 5347-5361, 2019.

Author Response

We are grateful for the review of our manuscript and its positive evaluation. In line with your remark, we have carefully traced the text and made the necessary corrections, e.g. lines 271, 276, 601. In the last paragraph of the introduction section (lines 98-100), we have introduced a suggested reference [27]. We hope that the changes we have introduced will find your approval.